# Using Ontology-Grounded Token Embeddings To Predict Prepositional Phrase Attachments

## Abstract

Type-level word embeddings use the same set of parameters to represent all instances of a word regardless of its context, ignoring the inherent lexical ambiguity in language. Instead, we embed semantic concepts (or synsets) as defined in WordNet and represent a word token in a particular context by estimating a distribution over relevant semantic concepts. We use the new, context-sensitive embeddings in a model for predicting prepositional phrase (PP) attachments and jointly learn the concept embeddings and model parameters. We show that using context-sensitive embeddings improves the accuracy of the PP attachment model by 5.4% absolute points, which amounts to a 34.4% relative reduction in errors.

## 1 Introduction

Type-level word embeddings map a word type (i.e., a surface form) to a dense vector of real numbers such that similar word types have similar embeddings. When pre-trained on a large corpus of unlabeled text, they provide an effective mechanism for generalizing statistical models to words which do not appear in the labeled training data for a downstream task.

In this paper, we make the following distinction between types and tokens: By word types, we mean the surface form of the word, whereas by tokens we mean the instantiation of the surface form in a context. For example, the same word type *'pool'* occurs as two different tokens in the sentences *"He sat by the pool,"* and *"He played a game of pool."*

Most word embedding models define a single vector for each word type. However, a fundamen-

tal flaw in this design is their inability to distinguish between different meanings and abstractions of the same word. In the two sentences shown above, the word *'pool'* has different meanings, but the same representation is typically used for both of them. Similarly, the fact that *'pool'* and *'lake'* are both kinds of water bodies is not explicitly incorporated in most type-level embeddings. Furthermore, it has become a standard practice to tune pre-trained word embeddings as model parameters during training for an NLP task (e.g., Chen and Manning, 2014; Lample et al., 2016), potentially allowing the parameters of a frequent word in the labeled training data to drift away from related but rare words in the embedding space.

Previous work partially addresses these problems by estimating concept embeddings in Word-Net (e.g., Rothe and Schütze, 2015), or improving word representations using information from knowledge graphs (e.g., Faruqui et al., 2015). However, it is still not clear how to use a lexical ontology to derive context-sensitive token embeddings.

In this paper, we represent a word token in a given context by estimating a context-sensitive probability distribution over relevant concepts in WordNet (Miller, 1995) and use the expected value (i.e., weighted sum) of the concept embeddings as the token representation (see §2). In addition to providing context-sensitive token embeddings, the proposed method implicitly regularizes the embeddings of related words by forcing related words to share similar concept embeddings. As a result, the representation of a rare word which does not appear in the training data for a downstream task benefits from all the updates to related words which share one or more concept embeddings.

Our approach to context-sensitive embeddings assumes the availability of a lexical ontology such

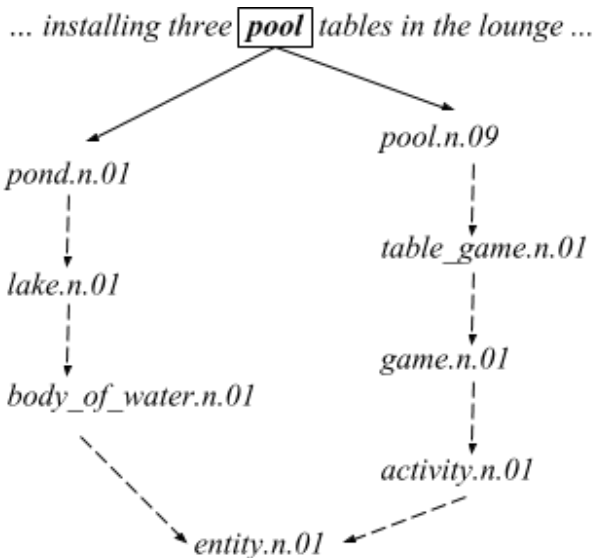

*... installing three* **pool** *tables in the lounge ...*

Figure 1: An example grounding for the word *'pool'*. Solid arrows represent possible senses and dashed arrows represent hypernym relations. Note that the same set of concepts are used to ground the word *'pool'* regardless of its context. Other WordNet senses for *'pool'* were removed from the figure for simplicity.

as WordNet, but it does not require a tagger for word senses. We use the proposed embeddings to predict prepositional phrase (PP) attachments (see §3), a challenging problem which emphasizes the selectional preferences between words in the PP and each of the candidate head words. Our empirical results and detailed analysis (see §4) show that the proposed embeddings effectively use WordNet to improve the accuracy of PP attachment predictions.

## 2 WordNet-Grounded Context-Sensitive Token Embeddings

In this section, we focus on defining our context-sensitive token embeddings. We first describe our grounding of word types using WordNet concepts. Then, we describe our model of context-sensitive token-level embeddings as a weighted sum of WordNet concept embeddings.

### 2.1 WordNet Grounding

We use WordNet to map each word type to a set of synsets, including possible generalizations or abstractions. Among the labeled relations defined in WordNet between different synsets, we focus on the hypernymy relation to help model generaliza-

tion and selectional preferences between words, which is especially important for predicting PP attachments (Resnik, 1993). To ground a word type, we identify the the set of (direct and indirect) hypernyms of the WordNet senses of that word. A simplified grounding of the word 'pool' is illustrated in Figure 1. This grounding is key to our model of token embeddings, to be described in the following subsections.

### 2.2 Context-Sensitive Token Embeddings

Our goal is to define a context-sensitive model of token embeddings which can be used as a drop-in replacement for traditional type-level word embeddings.

**Notation.** Let $Senses(w)$ be the list of synsets defined as possible word senses of a given word type $w$ in WordNet, and $Hypernyms(s)$ be the list of hypernyms for a synset $s$.[1] For example, according to Figure 1:

$$Senses(\text{pool}) = [\text{pond.n.01, pool.n.09}], \text{ and}$$
$$Hypernyms(\text{pond.n.01}) = [\text{pond.n.01, lake.n.01,}$$
$$\text{body\_of\_water.n.01, entity.n.01}] \quad (1)$$

Each WordNet synset $s$ is associated with a set of parameters $\mathbf{v}_s \in \mathbb{R}^n$ which represent its embedding. This parameterization is similar to that of Rothe and Schütze (2015).

**Embedding model.** Given a sequence of tokens $\boldsymbol{t}$ and their corresponding word types $\boldsymbol{w}$, let $\mathbf{u}_i \in \mathbb{R}^n$ be the embedding of the word token $t_i$ at position $i$. Unlike most embedding models, the token embeddings $\mathbf{u}_i$ are not parameters. Rather, $\mathbf{u}_i$ is computed as the expected value of concept embeddings used to ground the word type $w_i$ corresponding to the token $t_i$:

$$\mathbf{u}_i = \sum_{s \in Senses(w_i)} \sum_{s' \in Hypernyms(s)} p(s, s' \mid \boldsymbol{t}, \boldsymbol{w}, i) \, \mathbf{v}_{\mathbf{s'}} \quad (2)$$

such that

$$\sum_{s \in Senses(w_i)} \sum_{s' \in Hypernyms(s)} p(s', s \mid \boldsymbol{t}, \boldsymbol{w}, i) = 1 \quad (3)$$

---

[1]For notational convenience, we assume that $s \in Hypernyms(s)$.

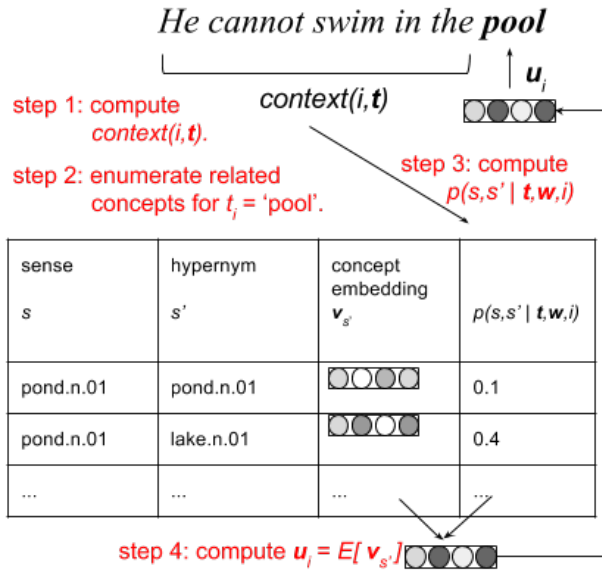

Figure 2: Steps for computing the context-sensitive token embedding for the word *'pool'*, as described in §2.2.

The distribution which governs the expectation over synset embeddings factorizes into two components:

$$p(s, s' \mid \boldsymbol{t}, \boldsymbol{w}, i) \propto \lambda_{w_i} \exp^{-\lambda_{w_i} \ rank(s, w_i)} \times$$
$$MLP([\mathbf{v}_{s'}; context(i, \boldsymbol{t})]) \quad (4)$$

The first component, $\lambda_{w_i} \exp^{-\lambda_{w_i} \ rank(s, w_i)}$, is a sense prior which reflects the prominence of each word sense for a given word type. Note that this is defined for each word type ($w_i$), and is shared across all tokens which have the same word type. The parameterization of the sense prior is similar to an exponential distribution since WordNet senses are organized in descending order of their frequency. The scalar parameter ($\lambda_{w_i}$) controls the decay of the probability mass.

The second component, $MLP([\mathbf{v}_{s'}; context(i, \boldsymbol{t})])$ is what makes the token representations context-sensitive. It scores each concept in the WordNet grounding of $w_i$ by feeding the concatenation of the concept embedding and a dense vector that summarizes the textual context into a multilayer perceptron (*MLP*) with two $tanh$ layers followed by a $softmax$ layer. This component is inspired by the soft attention often used in neural machine translation (Bahdanau et al., 2014).[2] The definition of the

———
[2]Although soft attention mechanism is typically used to

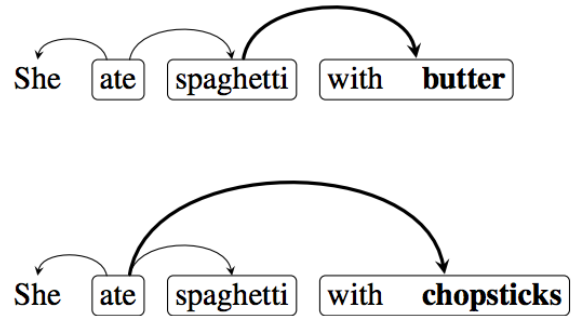

Figure 3: Two sentences illustrating the importance of lexicalization in PP attachment decisions. In the top sentence, the PP *'with butter'* attaches to the noun *'spaghetti'*. In the bottom sentence, the PP *'with chopsticks'* attaches to the verb *'ate'*. **Note:** This figure and caption have been reproduced from Belinkov et al. (2014).

*context* function is dependent on the encoder used to encode the context. We describe a specific instantiation of this function in §3.

To summarize, Figure 2 illustrates how to compute the embedding of a word token $t_i = \ 'pool'$ in a given context:

1. compute a summary of the context $context(i, \boldsymbol{t})$,

2. enumerate related concepts for $t_i$,

3. compute $p(s, s' \mid \boldsymbol{t}, \boldsymbol{w}, i)$ for each pair $(s, s')$, and

4. compute $\mathbf{u}_i = \mathbb{E}[\mathbf{v}_{s'}]$.

In the following section, we describe our model for predicting PP attachments, including our definition for *context*.

## 3 PP Attachment

Disambiguating PP attachments is an important and challenging NLP problem, and is a good fit for evaluating our WordNet-grounded context-sensitive embeddings since modeling hypernymy and selectional preferences is critical for successful prediction of PP attachments (Resnik, 1993).

Figure 3, reproduced from Belinkov et al. (2014), illustrates an example of the PP attachment prediction problem. The accuracy of a

———
explicitly represent the importance of each item in a sequence, it can also be applied to non-sequential items.

competitive English dependency parser at predicting the head word of an ambiguous prepositional phrase is 88.5%, significantly lower than the overall unlabeled attachment accuracy of the same parser (94.2%).[3]

This section formally defines the problem of PP attachment disambiguation, describes our baseline model, then shows how to integrate the token-level embeddings in the model.

### 3.1 Problem definition

We follow Belinkov et al. (2014)'s definition of the PP attachment problem. Given a preposition $p$ and its direct dependent $d$ in the prepositional phrase (PP), our goal is to predict the correct head word for the PP among an ordered list of candidate head words $\boldsymbol{h}$. Each example in the train, validation, and test sets consists of an input tuple $\langle \boldsymbol{h}, p, d \rangle$ and an output index $k$ to identify the correct head among the candidates in $\boldsymbol{h}$.

### 3.2 Model definition

Both our proposed and baseline models for PP attachment use bidirectional RNN with LSTM cells (bi-LSTM) to encode the sequence $\boldsymbol{t} = \langle h_1, \ldots, h_K, p, d \rangle$.

We score each candidate head by feeding the concatenation of the output bi-LSTM vectors for the head $h_k$, the preposition $p$ and the direct dependent $d$ through an MLP, with a fully connected *tanh* layer to obtain a non-linear projection of the concatenation, followed by a fully-connected *softmax* layer:

$$
\begin{aligned}
p(h_k \text{is head}) = MLP_{attach}([lstm\_out(h_k); \\
lstm\_out(p); \\
lstm\_out(d)]) \quad (5)
\end{aligned}
$$

To train the model, we use cross-entropy loss at the output layer for each candidate head in the training set. At test time, we predict the candidate head with the highest probability according to the model in Eq. 5, i.e.,

$$
\hat{k} = \arg\max_k p(h_k \text{is head} = 1). \quad (6)
$$

This model is inspired by the Head-Prep-Child-Ternary model of Belinkov et al. (2014). The main difference is that we replace the input features for each token with the output bi-RNN vectors.

---

[3]See Table 2 in §4 for detailed results.

We now describe the difference between the proposed model and the baseline. Generally, let $lstm\_in(t_i)$ and $lstm\_out(t_i)$ represent the input and output vectors of the bi-LSTM for each token $t_i \in \boldsymbol{t}$ in the sequence.

**Baseline model.** In the baseline model, we use type-level word embeddings to represent the input vector $lstm\_in(t_i)$ for a token $t_i$ in the sequence. The word embedding parameters are initialized with pre-trained vectors, then tuned along with the parameters of the bi-LSTM and $MLP_{attach}$. We call this model **LSTM-PP**.

**Proposed model.** In the proposed model, we use token level word embedding as described in §2 as the input to the bi-LSTM, i.e., $lstm\_in(t_i) = \mathbf{u}_i$. The context used for the attention component is simply the hidden state from the previous timestep. However, since we use a bi-LSTM, the model essentially has two RNNs, and accordingly we have two context vectors, and associated attentions. That is, $context_f(i, \boldsymbol{t}) = lstm\_in(t_{i-1})$ for the forward RNN and $context_b(i, \boldsymbol{t}) = lstm\_in(t_{i+1})$ for the backward RNN. The synset embedding parameters are initialized with pre-trained vectors and tuned along with the sense decay ($\lambda_w$) and MLP parameters from Eq. 4, the parameters of the bi-LSTM and those of $MLP_{attach}$. We call this model **OntoLSTM-PP**.

## 4 Experiments

**Dataset and evaluation.** We used the English PP attachment dataset created and made available by Belinkov et al. (2014). The training and test splits contain 33,359 and 1951 labeled examples respectively. As explained in §3.1, the input for each example is 1) an ordered list of candidate head words, 2) the preposition, and 3) the direct dependent of the preposition. The head words are either nouns or verbs and the dependent is always a noun. All examples in this dataset have at least two candidate head words. As discussed in Belinkov et al. (2014), this dataset is a more realistic PP attachment task than the RRR dataset (Ratnaparkhi et al., 1994). The RRR dataset is a binary classification task with exactly two head word candidates in all examples. The context for each example in the RRR dataset is also limited which defeats the purpose of our context-sensitive embeddings.

**Model specifications and hyperparameters.** For efficient implementation, we use mini-batch

updates with the same number of senses and hypernyms for all examples, padding zeros and truncating senses and hypernyms as needed. For each word type, we use a maximum of $S$ senses and $H$ indirect hypernyms from WordNet. In our initial experiments on a held-out development set (10% of the training data), we found that values greater than $S = 3$ and $H = 5$ did not improve performance. We also used the development set to tune the number of layers in $MLP_{attach}$ separately for the OntoLSTM-PP and LSTM-PP, and the number of layers in the attention MLP in OntoLSTM-PP. When a synset has multiple hypernym paths, we use the shortest one. Finally, words types which do not appear in WordNet are assumed to have one unique sense per word type with no hypernyms. Since the POS tag for each word is included in the dataset, we exclude WordNet synsets which are incompatible with the POS tag. The synset embedding parameters are initilized using the synset vectors obtained by running AutoExtend (Rothe and Schütze, 2015) on 100-dimensional GloVe (Pennington et al., 2014) vectors for WordNet 3.1. Representation for the OOV word types in LSTM-PP and OOV synset types in OntoLSTM-PP were randomly drawn from a uniform 100-d distribution. Initial sense prior parameters ($\lambda_w$) were also drawn from a uniform 1-d distribution.

**Baselines.** In our experiments, we compare our proposed model, OntoLSTM-PP with three baselines – LSTM-PP initialized with GloVe embedding, LSTM-PP initialized with GloVe vectors retrofitted to WordNet using the approach of Faruqui et al. (2015), and finally the best performing standalone PP attachment system from Belinkov et al. (2014), referred to as "HPCD (full)" in the paper. "HPCD (full)" is a neural network model that learns to compose the vector representations of each of the candidate heads with those of the preposition and the dependent, and predict attachments. The input representations are enriched using syntactic context information, POS, WordNet and VerbNet (Kipper et al., 2008) information and the distance of the head word from the PP is explicitly encoded in composition architecture. In contrast, we do not use syntactic context, VerbNet and distance information, and do not explicitly encode POS information.

| System | Initialization | Test Acc. |
|---|---|---|
| HPCD (full) | Skipgram | 88.7 |
| LSTM-PP | GloVe | 84.3 |
| LSTM-PP | GloVe-retro | 84.8 |
| OntoLSTM-PP | GloVe-extended | 89.7 |

Table 1: Results on Belinkov et al. (2014)'s PPA test set. HPCD (full) is from the original paper.

### 4.1 PP Attachment Results

Table 1 shows that our proposed token level embedding scheme "OntoLSTM-PP" outperforms the better variant of our baseline "LSTM-PP" (with GloVe-retro intialization) by an absolute accuracy difference of 4.9%, or a relative error reduction of 32%. "OntoLSTM-PP' also outperforms "HPCD (full)", the previous best result on this dataset.

Initializing the word embeddings with *GloVe-retro* (which uses WordNet as described in Faruqui et al. (2015)) instead of "textitGloVe" amounts to a small improvement, compared to the improvements obtained using "OntoLSTM-PP". This result illustrates that our approach of dynamically choosing a context sensitive distribution over synsets is a more effective way of making use of WordNet.

**Effect on dependency parsing.** Following Belinkov et al. (2014), we used RBG parser (Lei et al., 2014), and modified it by adding a binary feature indicating the PP attachment predictions from our model.

We compare four ways to compute the additional binary features: 1) the predictions of the best standalone system "HPCD (full)" in Belinkov et al. (2014), 2) the predictions of our baseline model "LSTM-PP", 3) the predictions of our improved model "OntoLSTM-PP", and 4) the gold labels "Oracle PP".

Table 2 shows the effect of using the PP attachment predictions as features within a dependency parser. We note there is a relatively small difference in unlabeled attachment accuracy for all dependencies (not only PP attachments), even when gold PP attachments are used as additional features to the parser. However, when gold PP attachment are used, we note a large potential improvement of 10.46 points (between the PPA accuracy for "RBG" and "RBG + Oracle PP"), which confirms that adding PP predictions as features is

| System | Full UAS | PPA Acc. |
|--------|----------|----------|
| RBG | 94.17 | 88.51 |
| RBG + HPCD (full) | 94.19 | 89.59 |
| RBG + LSTM-PP | 94.14 | 86.35 |
| RBG + OntoLSTM-PP | 94.30 | 90.11 |
| RBG + Oracle PP | 94.60 | 98.97 |

Table 2: Results from RBG dependency parser with features coming from various PP attachment predictors and oracle attachments.

an effective approach. Our proposed model "RBG + OntoLSTM-PP" recovers 15% of this potential improvement, while "RBG + HPCD (full)" recovers 10%, which illustrates that PP attachment remains a difficult problem with plenty of room for improvements even when using a dedicated model to predict PP attachments and using its predictions in a dependency parser.

We also note that, although we use the same predictions of the "HPCD (full)" model in Belinkov et al. (2014),[4] we report different results than Belinkov et al. (2014). For example, the unlabeled attachment score (UAS) of the baselines "RBG" and "RBG + HPCD (full)" are 94.17 and 94.19, respectively, in Table 2, compared to 93.96 and 94.05, respectively, in Belinkov et al. (2014). This is due to the use of different versions of the RBG parser.[5]

### 4.2 Analysis

In this subsection, we analyze different aspects of our model in order to develop a better understanding of its behavior.

**Effect of context sensitivity and sense priors.** We now show some results that indicate the relative strengths of two components of our context-sensitive token embedding model. The second row in Table 3 shows the test accuracy of a system trained without sense priors (that is, making $p(s|w_i)$ from Eq. 2 a uniform distribution), and the third row shows the effect of making the token representations context-insensitive by giving a similar attention score to all related concepts, essentially making them type level representations,

[4]The authors kindly provided their predictions for 1942 test examples (out of 1951 examples in the full test set). In Table 2, we use the same subset of 1942 test examples and will include a link to the subset in the final draft.

[5]We use the latest commit (SHA: e07f74dd2ba47348fd548935155ded38eea20809) on the GitHub repository of the RGB parser.

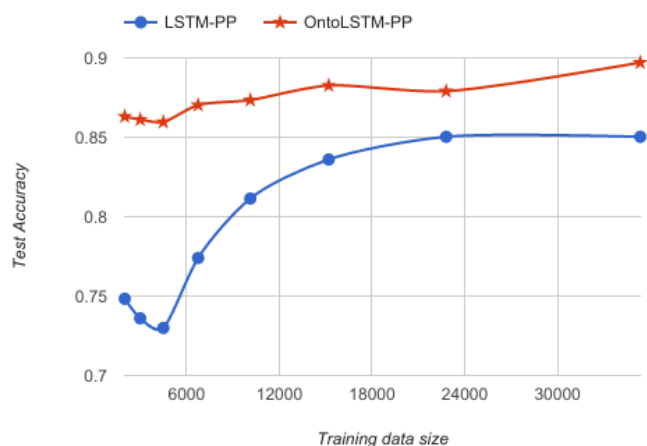

Figure 4: Effect of training data size on test accuracies of OntoLSTM-PP and LSTM-PP

but still grounded in WordNet. As it can be seen, removing context sensitivity has an adverse effect on the results. This illustrates the importance of the sense priors and the attention mechanism.

It is interesting that, even without sense priors and attention, the results with WordNet grounding is still higher than that of the two LSTM-PP systems in Table 1. This result illustrates the regularization behavior of sharing concept embeddings across multiple words, which is especially important for rare words.

**Effect of training data size.** Since "OntoLSTM-PP" uses external information, the gap between the model and "LSTM-PP" is expected to be more pronounced when the training data sizes are smaller. To test this hypothesis, we trained the two models with different amounts of training data and measured their accuracies on the test set. The plot is shown in Figure 4. As expected, the gap tends to be larger at lower data sizes. Surprisingly, even with 2000 sentences in the training data set, OntoLSTM-PP outperforms LSTM-PP trained with the full data set. When both the models are trained with the fill dataset, LSTM-PP reaches a training accuracy of 95.3%, whereas OntoLSTM-PP reaches 93.5%. The fact that LSTM-PP is overfitting the training data more, indicates the regularization capability of OntoLSTM-PP.

**Qualitative analysis.** To better understand the effect of WordNet grounding, we took a sample of sentences from the test set whose PP attachments were correctly predicted by OntoLSTM-PP but not by LSTM-PP. A common pattern observed was

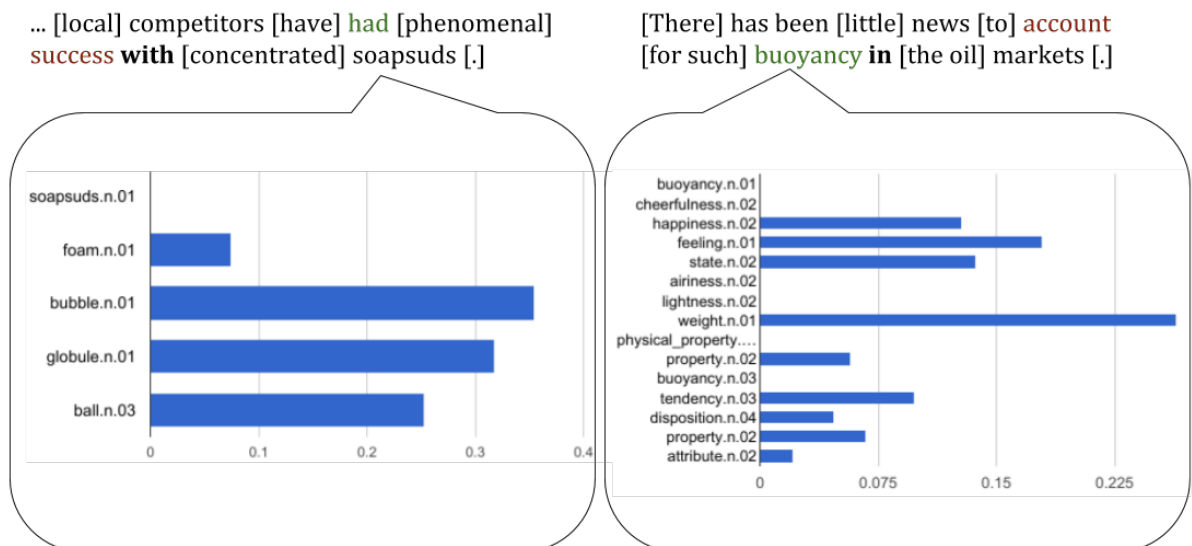

Figure 5: Two examples from the test set where OntoLSTM-PP predicts the head correctly and LSTM-PP does not, along with weights by OntoLSTM-PP to synsets that contribute to token representations of infrequent word types. The prepositions are shown in bold, LSTM-PP's predictions in red and OntoLSTM-PP's predictions in green. Words that are not candidate heads or dependents are shown in brackets.

| Model | PPA Acc. |
|---|---|
| full | 89.7 |
| - sense priors | 88.4 |
| - attention | 87.5 |

Table 3: Effect of removing sense priors and context sensitivity (attention) from the model

that those sentences contained words not seen frequently in the training data. Figure 5 shows two such cases. In both cases, the weights assigned by OntoLSTM-PP to infrequent words are also shown. The word types *soapsuds* and *buoyancy* do not occur in the training data, but OntoLSTM-PP was able to leverage the parameters learned for the synsets that contributed to their token representations. Another important observation is that the word type *buoyancy* has four senses in Word-Net (we consider the first three), none of which is the metaphorical sense that is applicable to *markets* as shown in the example here. Selecting a combination of relevant hypernyms from various senses may have helped OntoLSTM-PP make the right prediction. This shows the value of using hypernymy information from WordNet. Moreover, this indicates the strength of the hybrid nature of the model, that lets it augment ontological information with distributional information.

**Parameter space** We note that the vocabulary sizes in OntoLSTM-PP and LSTM-PP are comparable as the synset types are shared across word types. In our experiments with the full PP attachment dataset, we learned embeddings for 18k synset types with OntoLSTM-PP and 11k word types with LSTM-PP. Since the biggest contribution to the parameter space comes from the embedding layer, the complexities of both the models are comparable.

**Implementation and code availability.** The models are implemented using Keras (Chollet, 2015), and the functionality is available in the form of Keras layers to make it easier for other researchers to use the proposed embeddingn model.

**Future work.** This approach may be extended to other NLP tasks that can benefit from using encoders that can access WordNet information. WordNet also has some drawbacks, and may not always have sufficient coverage given the task at hand. As we have shown in §4.2, our model can deal with missing WordNet information by augmenting it with distributional information. Moreover, the methods described in this paper can be extended to other kinds of structured knowledge sources like Freebase which may be more suitable for tasks like question answering.

## 5 Related Work

This work is related to various lines of research within the NLP community: dealing with synonymy and homonymy in word representations both in the context of distributed embeddings and more traditional vector spaces; hybrid models of distributional and knowledge based semantics; and selectional preferences and their relation with syntactic and semantic relations.

The need for going beyond a single vector per word-type has been well established for a while, and many efforts were focused on building multi-prototype vector space models of meaning (Reisinger and Mooney, 2010; Huang et al., 2012; Chen et al., 2014; Jauhar et al., 2015; Neelakantan et al., 2015, etc.). However, the target of all these approaches is obtaining multi-sense word vector spaces, either by incorporating sense tagged information or other kinds of external context. The number of vectors learned is still fixed, based on the preset number of senses. In contrast, our focus is on learning a context dependent distribution over those concept representations. Other work not necessarily related to multi-sense vectors, but still related to our work includes Belanger and Kakade (2015)'s work which proposed a Gaussian linear dynamical system for estimating token-level word embeddings, and Vilnis and McCallum (2014)'s work which proposed mapping each word type to a density instead of a point in a space to account for uncertainty in meaning. These approaches do not make use of lexical ontologies and is not amenable for joint training with a downstream NLP task.

Related to the idea of concept embeddings is Rothe and Schütze (2015) who estimated Word-Net synset representations, given pre-trained type-level word embeddings. In contrast, our work focuses on estimating token-level word embeddings as context sensitive distributions of concept embeddings.

There is a large body of work that tried to improve word embeddings using external resources. Yu and Dredze (2014) extended the CBOW model (Mikolov et al., 2013) by adding an extra term in the training objective for generating words conditioned on similar words according to a lexicon. Jauhar et al. (2015) extended the skipgram model (Mikolov et al., 2013) by representing word senses as latent variables in the generation process, and used a structured prior based on the ontology.

Faruqui et al. (2015) used belief propagation to update pre-trained word embeddings on a graph that encodes lexical relationships in the ontology. In contrast to previous work that was aimed at improving *type level* word representations, we propose an approach for obtaining *context-sensitive* embeddings at the *token level*, while jointly optimizing the model parameters for the NLP task of interest.

Resnik (1993) showed the applicability of semantic classes and selectional preferences to resolving syntactic ambiguity. Zapirain et al. (2013) applied models of selectional preferences automatically learned from WordNet and distributional information, to the problem of semantic role labeling. Resnik (1993); Brill and Resnik (1994); Agirre (2008) and others have used WordNet information towards improving prepositional phrase attachment predictions.

## 6 Conclusion

In this paper, we proposed a grounding of lexical items which acknowledges the semantic ambiguity of word types using WordNet and a method to learn a context-sensitive distribution over their representations. We also showed how to integrate the proposed representation with recurrent neural networks for disambiguating prepositional phrase attachments, showing that the proposed WordNet-grounded context-sensitive token embeddings outperforms standard type-level embeddings for predicting PP attachments. We provided a detailed qualitative and quantitative analysis of the proposed model.

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
