# Peer review of "Ontology-Aware Token Embeddings for Prepositional Phrase Attachment"

_ACL 2017 — decision unknown_

[Official Review · Reviewer 1 · rating 2 · confidence 5]
soundness 3 · originality 4 · clarity 2 · impact 5 · substance 4 · appropriateness 5 · meaningful comparison 5 · presentation format Poster

- Overview:

The paper proposes a new model for training sense embeddings grounded in a
lexical-semantic resource (in this case WordNet). There is no direct evaluation
that the learned sense vectors are meaningful; instead, the sense vectors are
combined back into word embeddings, which are evaluated in a downstream task:
PP attachment prediction.

- Strengths:

PP attachment results seem solid.

- Weaknesses:

Whether the sense embeddings are meaningful remains uninvestigated. 

The probabilistic model has some details that are hard to understand. Are the
\lambda_w_i hyperparameters or trained? Where does “rank” come from, is
this taken from the sense ranks in WordNet?

Related work: the idea of expressing embeddings of words as a convex
combination of sense embeddings has been proposed a number of times previously.
For instance, Johansson and Nieto Piña “Embedding a semantic network in a
word space” (NAACL, 2015) decomposed word embeddings into ontology-grounded
sense embeddings based on this idea. Also in unsupervised sense vector training
this idea has been used, for instance by Arora et al “Linear Algebraic
Structure of Word Senses, with Applications to Polysemy”.

Minor comments:

no need to define types and tokens, this is standard terminology

why is the first \lamba_w_i in equation 4 needed if the probability is
unnormalized?

- General Discussion: